# Plant invasion and naturalization are influenced by genome size, ecology and economic use globally

Kun Guo [1,2], Petr Pyšek [3,4], Mark van Kleunen [5,6], Nicole L. Kinlock[5], Magdalena Lučanová [7,8], Ilia J. Leitch [9], Simon Pierce[10], Wayne Dawson [11,12], Franz Essl[13], Holger Kreft [14,15,16], Bernd Lenzner [13], Jan Pergl[3], Patrick Weigelt [14,15,16] & Wen-Yong Guo [1,2,17] ✉

Human factors and plant characteristics are important drivers of plant invasions, which threaten ecosystem integrity, biodiversity and human well-being. However, while previous studies often examined a limited number of factors or focused on a specific invasion stage (e.g., naturalization) for specific regions, a multi-factor and multi-stage analysis at the global scale is lacking. Here, we employ a multi-level framework to investigate the interplay between plant characteristics (genome size, Grime's adaptive CSR-strategies and native range size) and economic use and how these factors collectively affect plant naturalization and invasion success worldwide. While our findings derived from structural equation models highlight the substantial contribution of human assistance in both the naturalization and spread of invasive plants, we also uncovered the pivotal role of species' adaptive strategies among the factors studied, and the significantly varying influence of these factors across invasion stages. We further revealed that the effects of genome size on plant invasions were partially mediated by species adaptive strategies and native range size. Our study provides insights into the complex and dynamic process of plant invasions and identifies its key drivers worldwide.

Biological invasions cause extensive ecological impacts by increasing the risk of biodiversity loss, especially of native species[1–3], and by redefining biogeographical boundaries[4,5]. Invasive species also affect human well-being and cause substantial economic losses globally[6–8]. With the world facing growing anthropogenic pressures and becoming increasingly interconnected, projections indicate a surge in the number of alien species in future decades[7,9,10]. Therefore, disentangling the factors that determine variation in species' invasion success is of great importance to both basic ecology and conservation efforts.

The compilation of data on the global distribution of alien plant species has been instrumental in identifying key factors that drive plant invasions, including human-associated factors and species characteristics (Fig. 1). For instance, a recent study found that the likelihood of global naturalization is 18 times higher for plants with economic uses (e.g., horticulture, human food, animal fodder, medicines) compared to species with no known economic use (Fig. 1; path 11)[11]. This finding likely stems from the fact that economically useful alien plants often have higher propagule pressure (i.e., the number of propagules entering a new region). This then increases the likelihood that they can overcome barriers related to demographic, environmental, and genetic stochasticity, thus enhancing their success as invaders[12,13].

The roles of species' characteristics, such as functional traits and native geographic range size, in influencing the success of plant invasions have also been extensively explored (Fig. 1; paths 1, 9, and 10). For example, studies have shown that plants with large genomes (i.e., the

**Fig. 1 | Conceptual framework showing relationships between some well-validated factors contributing to plant invasions, i.e., genome size, specific functional traits or general adaptive strategy i.e., Grime's CSR strategy, native range size, and economic use.** Blue, red, black, and grey arrows correspond to positive, negative, variable and unknown associations of the paths, as shown by the cited studies in Table 1. Descriptions and relevant references for each path are also discussed in the introduction. In Grime's CSR (C–competitor; S–stress-tolerator; R–ruderal) adaptive strategy theory, competitors have efficient resource acquisition and allocation strategies, stress-tolerators allocate resources towards stress resistance rather than rapid growth, whereas ruderals invest their resources in producing abundant seeds and establishing new individuals[31,32]. Icons representing functional traits were adapted from pictograms courtesy of PhyloPic (www.phylopic.org).

amount of nuclear DNA) are less likely to naturalize compared to those with smaller genomes[14–17], providing support for the "large genome constraint" hypothesis[18]. Larger genomes tend to result in increased cell sizes and decreased cell-division rates, subsequently affecting organism-level functional traits (Fig. 1; path 3)[19]. Thus, plants with small genomes can exhibit significant variation in seed size, whereas plants with large genomes are typically restricted to having large seeds[20,21]. Furthermore, plants with large genomes are typically obligate perennials, whereas those with small genomes adopt various life-cycle strategies[22,23]. Ultimately, these traits affect the habitat breadth and range size of a species and, consequently, its invasion potential[14,17,24,25]. Nevertheless, a small genome may not always be advantageous. For example, a small genome may constrain the invasive spread of species[17]. This may potentially be because of polyploidy in the larger-genome species, as polyploidy not only results in a step change in genome size (at least initially) but can also generate heterozygosity, which might enhance competitive ability and increase the likelihood of successful invasion into new environments[26].

Despite numerous attempts to identify key traits affecting plant invasions, consistent associations have proven elusive[27–29]. Studies have shown that the integration of sets of functional traits that represent adaptive strategies holds potential for explaining plant invasion success. For example, species with acquisitive growth strategies, characterized by a high leaf nitrogen content and low leaf mass per area, were found to be among the most successful invaders in Europe[30]. Recently, Grime's CSR strategy theory (C–competitor; S–stress-tolerator; R–ruderal)[31,32], framed within the context of the global spectrum of plant form and function (i.e., plant size and fast-slow economics)[33], offers a promising framework for explaining plant invasions. Research using this theory has revealed that worldwide, C-

and R-selected plants are more likely to naturalize than S-selected species (Fig. 1; path 10)[34,35]. Typically, competitors exhibit rapid growth and achieve large sizes, facilitating resource monopolization during competition[36]. Stress-tolerators possess dense and persistent tissues, enabling their survival in resource-poor and abiotically variable habitats. Ruderals allocate a large proportion of resources to propagules and prioritize fast completion of the life cycle to ensure reproduction before the next disturbance event[37]. It is highly likely that adaptive strategies are also related to genome size, as R-selected species typically produce many small seeds, while C-selected species tend to have larger seeds[38]. However, there have been no empirical tests validating these expectations as yet (Fig. 1; path 4).

Native range size is another important characteristic that exhibits a positive correlation with alien species' invasion success[39–41]. Compared to species with restricted native distributions, those with large native ranges have a higher possibility of establishing and spreading as alien species due to their preadaptation to a wide range of environmental conditions and/or high dispersal ability. For example, R-selected species characterized by traits that promote colonization, such as high fecundity and fast growth, tend to have large native ranges and are more likely to naturalize[34,42]. Additionally, plants with large native ranges are more likely to be used by humans[34,40,43], leading to increased propagule pressure and, ultimately, enhanced invasion success. These associations between native range size, species traits and human uses highlight the complex interactions among factors that convey plant invasion success.

It is well-known that different drivers play varying roles across the different invasion stages[40,44–47]. It follows that the above-mentioned factors may vary in their importance across the invasion continuum. The invasion continuum is a process that starts with human-mediated introductions to new areas, followed by the naturalization stage, in which self-sustaining, persistent populations are formed, and finally, the invasion stage is characterized by a rapid spread of the species across the landscape[48–51]. Previous studies aiming to disentangle drivers of species' invasion have typically focused on a single category of explanatory factors[52], such as species traits[28,53,54], or on a specific invasion stage, such as naturalization[11,34]. Although some studies have explored multiple invasion stages, they have mainly focused on regional floras[39,46,55–57]. These studies thus provide only a snapshot of the multi-faceted relationships among multiple potential drivers and invasion success at the regional scale, whereas a global-scale assessment is still missing.

Here, we compiled information on key species characteristics (genome size, Grime's adaptive CSR-strategies, and native range size) and economic use to explore how these factors interact with each other, both directly and indirectly, to shape the naturalization and invasion success of plants on a global scale (see Fig. 1). Specifically, we used Bayesian structural equation models (SEMs)[58,59] to assess how these factors affect multiple metrics of invasion success[17,60,61]: (i) naturalization incidence (whether a species has naturalized anywhere in the world), (ii) naturalization extent (in how many regions an alien species has naturalized, provided that it has naturalized in at least one region) and (iii) invasion extent (the number of regions where a naturalized species has been recorded as invasive). Both naturalization incidence and extent were modeled to capture different facets of the naturalization stage and to assess whether they have different underlying drivers. Additionally, our models account for phylogenetic relatedness among species to recognize the potential influence of evolutionary history on invasion success. Our study underscores the varying effects of different factors along the invasion continuum, the central role of Grime's CSR adaptive strategies among the tested factors and the dominant contribution of economic uses to plant invasions. Additionally, we show that the effects of genome size on the likelihood of a plant becoming invasive are partially mediated by other factors tested here.

**Table 1 | Detailed descriptions and key references corresponding to each of the 12 paths in the conceptual framework shown in Fig. 1**

| Path | Description | Relationships | Key references |
|---|---|---|---|
| 1 | Plants with larger genome sizes are less likely to succeed in invasion. | ↘ | 14,25 |
| 2 | Plants with larger genome sizes tend to have narrow native range sizes. | ↘ | 18 |
| 3 | Various relationships between plant genome size and a diversity of functional traits relevant to invasion success have been reported. | → | 15,25 |
| 4 | Relationships between plant genome size and adaptative strategies (i.e., Grime's CSR framework, Grime and Pierce 2012) remain unknown. | | |
| 5 | Relationships between plant genome size and their economic use remain unknown. | | |
| 6 | Various relationships between plant species' traits and their native range size are reported. | → | 29,55 |
| 7 | Plants with certain traits are more frequently used by humans. | → | 11,34 |
| 8 | Plants with wider native ranges are more frequently used by humans. | ↗ | 34,43 |
| 9 | Plants with wider native ranges are more likely to succeed in invasion. | ↗ | 34,40 |
| 10 | Various trait-invasion relationships are reported. | → | 27,28,34,35 |
| 11 | Plants which have economic uses are more likely to become naturalized and invasive. | ↗ | 11 |
| 12 | Grime's CSR strategies represent plants' overall adaptation to environmental conditions. | | 31,37 |

Up-right arrows, down-right arrows, and horizontal arrows, respectively, represent positive, negative and variable relationships.

## Results

We compiled a global dataset on native range size, genome size, economic use and CSR values of vascular plant species, and their geographic naturalization and invasiveness status (i.e., countries, federal states or provinces of large countries, islands). Note that data on both holoploid (amount of DNA in the nucleus) and monoploid (amount of DNA in one chromosome set) genome sizes were collected, but we mainly focus on the results from the holoploid datasets in the main text as they were similar to the results obtained for the monoploid genome-size set, which are in the Supplementary materials. For the holoploid dataset, complete data for a total of 1612 species were obtained. Among these, 419 species have no known naturalization occurrences, 1193 species were identified as naturalized alien species (Fig. 2), and 618 of the naturalized species were classified as invasive in at least one region worldwide. Further, most of the 1612 species are angiosperms ($n = 1545$), representing 115 families and 46 orders based on the Angiosperm Phylogeny Group classification[62]. Additionally, there were 32 gymnosperm species and 35 monilophyte species included in the dataset.

Structural equation models on naturalization incidence (Fig. 3a–c and Supplementary Fig. 1) revealed no significant direct association between holoploid genome size and naturalization incidence. However, there was a high probability that C-scores increased (estimate = 0.21, 95% credible interval, hereafter CI = [0.15, 0.27], $P_{est>0} = 1.000$) whilst S-scores (estimate = −0.08, CI = [−0.13, −0.02], $P_{est<0} = 0.981$), R-scores (estimate = −0.11, CI = [−0.17, −0.04], $P_{est<0} = 0.998$) and native range size (all $P_{est<0} \geq 0.998$) decreased with larger holoploid genome size. The results also revealed a high probability (all $P_{est>0} = 1.000$) of naturalization incidence increasing with higher C-scores (estimate = 0.40, CI = [0.24, 0.56]), greater number of economic uses, larger native range size and smaller S-scores (estimate = −0.34, CI = [−0.50, −0.19]).

For each of the three CSR scores, models of naturalization extent showed similar results to models of naturalization incidence, with consistent directions of associations (Fig. 3a–c vs. 3d–f; Supplementary Figs. 1 vs. 2). However, the magnitudes of the associations changed substantially. For instance, compared to CSR scores, native range size exhibited much stronger associations with naturalization incidence, whereas equivalent effects of native range size and CSR scores on naturalization extent were found (Fig. 3a vs. 3d). Furthermore, in contrast to the strong evidence of increased naturalization incidence with higher C-scores, both C-scores (estimate = 0.08, [0.03, 0.13],

$P_{est>0} = 0.996$) and R-scores (estimate = 0.15, [0.10, 0.19], $P_{est>0} = 1.000$) were positively related to naturalization extent.

Models of invasion extent (Fig. 3g–i and Supplementary Fig. 3) indicate that it increased with naturalization extent but decreased with an increasing R-score (estimate = −0.10, [−0.17, 0.03], $P_{est<0} = 0.992$). Additionally, we found a direct positive association between economic use and invasion extent (all $P_{est>0} \geq 0.996$), while the CSR scores were indirectly related to invasion extent through economic uses and/or naturalization extent (Fig. 3g–i).

The results from the holoploid dataset showed a direct relationship only between holoploid genome size and naturalization extent, whereas the results from the monoploid-focused dataset showed significant associations between monoploid genome size and all three metrics of plant invasions. Notably, SEMs on the three measures of plant invasion success from both datasets revealed several indirect paths from genome size to plant invasions via species characteristics (Supplementary Table 1) e.g., genome size → CSR scores and/or native range size → plant invasions.

## Discussion

By compiling a global dataset and integrating data on factors that have been shown to be related to invasion success, we explored how species characteristics and their usefulness for humans affects plant naturalization and invasion at the global scale. Our study shows a complex but clear pathway from genome size to invasion mediated by its influence on ecological strategies (represented using the CSR score), native range size (which encompasses numerous ecological/evolutionary relationships) and human behavior (represented as economic use), while accounting for alternative pathways.

Our results provide empirical evidence showing how the impacts of different factors, such as whether a species has an economic use and various species characteristics (e.g., range size, CSR categories), vary depending on the stage along the invasion continuum at the global scale. For example, R-scores were not significantly related to naturalization incidence but positively related to naturalization extent. Additionally, several other studies have also reported changing roles of factors along the invasion continuum. For example, Divíšek et al.[45] showed that similarity in functional traits (i.e., specific leaf area, plant height and seed weight) between introduced and native plant species facilitated naturalization success but inhibited invasion success. From an evolutionary perspective, Omer et al.[47] found that plant species that were phylogenetically distant from native flora were more likely to be

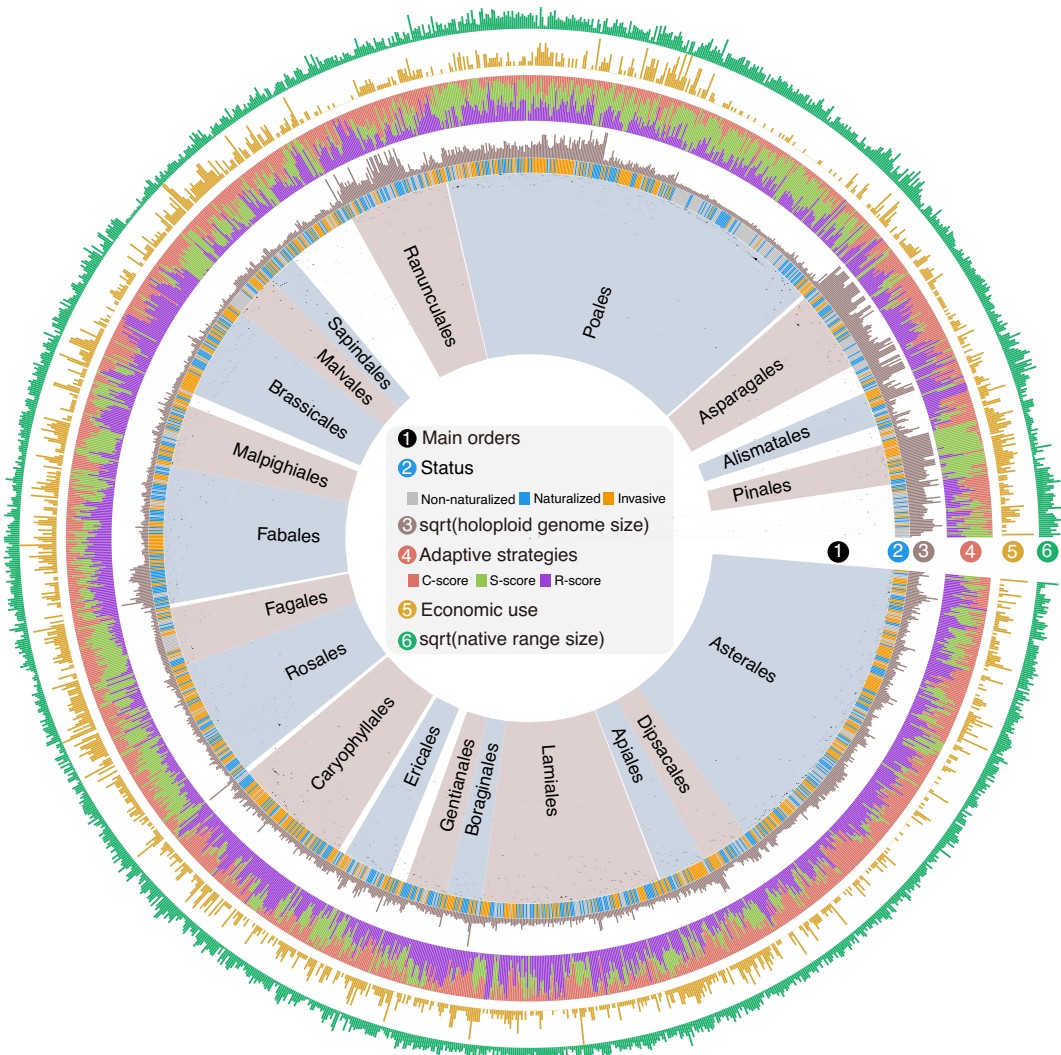

**Fig. 2 | Species status (non-naturalized, naturalized and invasive), holoploid genome size, CSR adaptive strategy scores, number of economic uses and native range size of the 1612 species projected on the phylogeny[89].** Names of the 20 largest orders in the phylogeny are shown in the innermost circle. To facilitate visualization, holoploid genome size and native range size were square-root transformed (sqrt).

introduced, whereas those closely related to natives were more likely to naturalize, and that naturalized species distantly related to natives had a higher probability of becoming invasive. These previous findings, together with the present study, emphasize the stage-dependent nature of plant invasion.

The "large genome constraint" hypothesis predicts a negative correlation between genome size and invasion success[14,17,63]. Our observed direct associations between both holoploid (Fig. 3 and Supplementary Figs. 1–3) and monoploid (Supplementary Figs. 4–7) genome size and different measures of plant invasion success support the predictions of this hypothesis. Moreover, while a recent study by Pyšek et al.[17] acknowledges the direct significant impact of genome size on plant naturalization and invasion, our SEMs at least partly unravel the underlying mechanisms of the large genome constraint effect by revealing two possible indirect pathways from genome size to plant invasion success (Supplementary Table 1). These indirect pathways involve: (i) species' characteristics, represented by the pathway genome size → adaptive strategies and/or native range size → plant invasion; and (ii) economic uses, represented by the pathway genome size → adaptive strategies and/or native range size → economic uses → plant invasion. It is not surprising that plant genome size plays a role in influencing species traits[25], and previous studies have

also reported connections among adaptive strategies, native range size, economic uses and plant invasions[11,34,42,64]. However, our study is the first to synthesize these associations with a global dataset using SEMs.

Our results further emphasize the dominant contribution of economic use in naturalization success[11]. Compared to the adaptive strategies, the number of economic uses identified for a species exhibited much stronger associations with naturalization incidence and extent. Similarly, Guo et al. showed that ornamental use (included in environmental use, one of the eight categories of economic uses in this study – see Supplementary Table 2) had much stronger effects than adaptive strategies on naturalization incidence and extent[34]. We also identified direct links between economic use and invasion extent, highlighting the persistent role of usefulness to humans throughout the invasion continuum. These results also corroborate that factors associated with globalization and economic growth are key drivers of biological invasions worldwide[65–67]. Apart from such direct associations between economic use and plant invasions, the above-mentioned indirect path from genome size to plant invasions via economic uses suggests complex interactions between species characteristics and economic uses. Nonetheless, apparent pathways can be discerned i.e., competitive species with widespread native ranges are more likely to

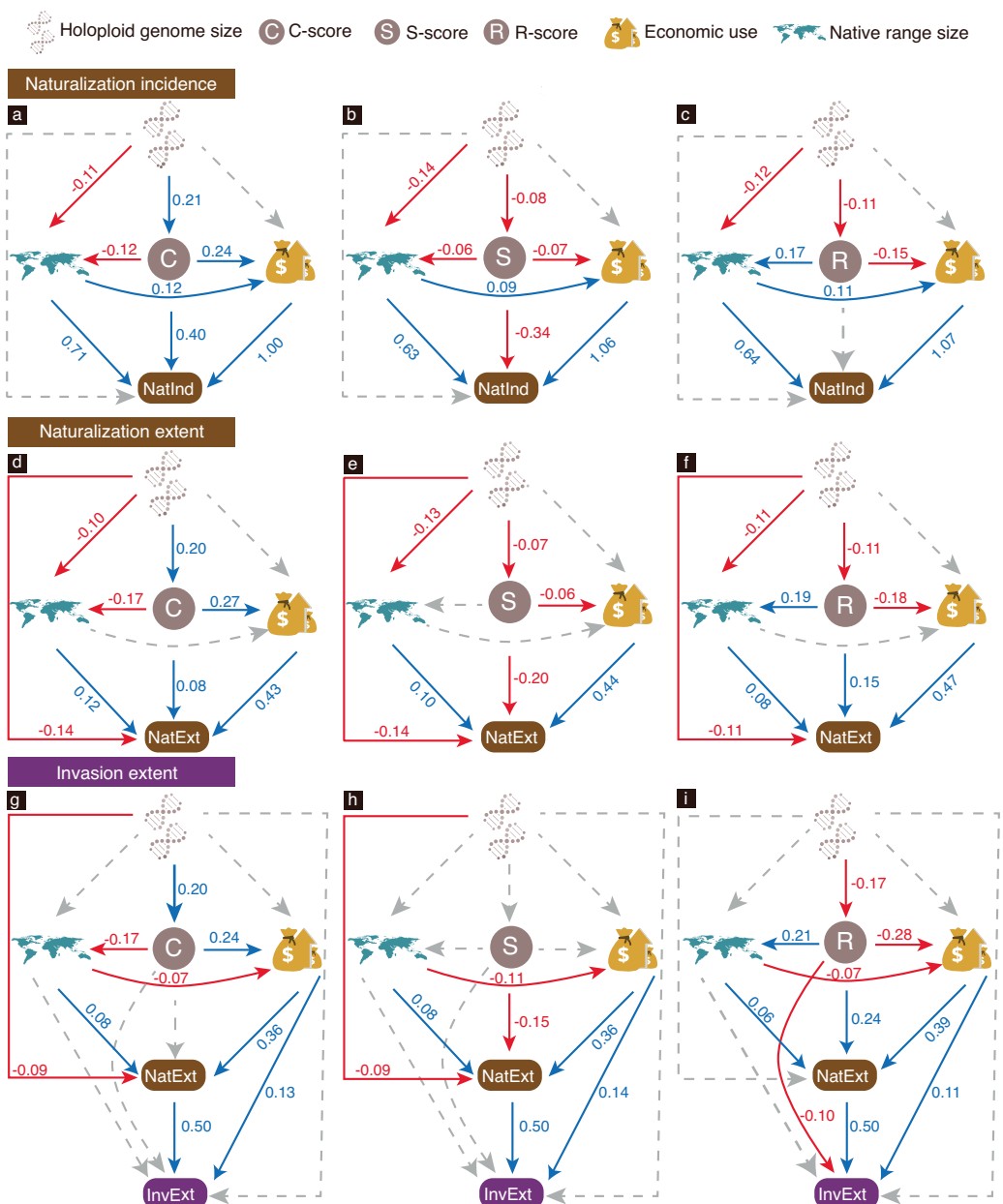

**Fig. 3 | Direct and indirect paths from holoploid genome size to plant invasion via species characteristics (i.e., native range size, Grime's CSR strategies, and economic use).** Structural equation models linking plant holoploid genome size, CSR-strategy scores, native range size, and economic use to **a**–**c** naturalization incidence (NatInd i.e., whether the species is known to be naturalized anywhere; $n = 1612$), **d**–**f** naturalization extent (NatExt i.e., the number of regions a species has been recorded in as naturalized; $n = 1193$), and **g**–**i** invasion extent (InvExt i.e., the number of regions a species has been recorded in as invasive; $n = 618$). Solid red lines, solid blue lines and dashed grey lines, respectively, indicate negative, positive and non-significant (95% credible interval includes zero) relationships. Numbers beside the arrows are standardized coefficients, which are only shown for significant paths. See Figs. S1–3 for details of each path.

be used economically (Fig. 3 and Supplementary Fig. 4). Moreover, as we employed an impact-based definition of invasion, our findings imply that economically valuable alien plants can be ecologically and possibly also economically harmful i.e., can have profound negative effects on native ecosystems and national economies.

Although a strength of our study is the utilization of various global datasets such as the Global Naturalized Alien Flora (GloNAF), it is important to acknowledge certain limitations of our dataset. Specifically, we lacked data that directly capture introduction effort and casual occurrences, meaning that possible filter effects of the early invasion stages are missing. In addition, due to the merging of data from different sources, only a small subset of the global flora, limited to 1612 vascular plants, has data on all factors encompassed in our

conceptual framework. The most limiting variables (in terms of number of species with data available) were genome size and adaptive strategy. Therefore, concerted efforts are needed to increase the availability of such data for larger numbers of species and other vascular plant lineages (i.e., gymnosperms, monilophytes and lycophytes).

Despite these limitations, our study represents a significant advance in the understanding of plant invasion dynamics by incorporating three key species characteristics and economic uses within a multi-factor and multi-stage framework. The results underscore the stage-dependent nature of plant invasions and the usefulness of adaptive strategies in elucidating the underlying mechanisms. Additionally, we contribute to the understanding of genome-invasion

relationships by identifying indirect pathways linking genome size to plant invasions through species characteristics and economic uses. Specifically, our results demonstrate that species with larger genomes (which most likely arose through polyploidization) are more likely to be C-selected and less likely to be S- or R-selected species. Finally, our study emphasizes the importance of including factors reflecting human assistance (e.g., economic uses) in gaining an improved understanding of plant invasions.

## Methods

### CSR scores and genome size

We compiled CSR scores for 4151 species worldwide. CSR scores were quantified based on three traits demonstrated to strongly represent the principle functional space of plants: leaf area (LA; representing the plant/organ size spectrum), specific leaf area (SLA; high values representing 'acquisitive' plant resource economics), and leaf dry matter content (LDMC; high values representing 'conservative' economics)[31]. While only three traits suffice for CSR calculation, they also exhibit significant statistical correlations with a more extensive range of plant characteristics, encompassing whole plant traits (canopy height, lateral spread), leaf traits (leaf nitrogen and carbon concentrations), and reproductive traits (seed mass, seed volume, seed variance, total mass of seeds, flowering period and flowering start) in the world flora (see Pierce et al.[31] for a multivariate analysis of these relationships).

Data for these three traits were collated from multiple sources[34,68–72]. In instances where multiple trait values were available for a species, we used the mean values for the CSR calculation. The 'StrateFy' CSR classification tool of Pierce et al.[31] employed here does not simply use each trait to directly represent each axis. Instead, it determines the trade-off between traits (i.e., increased values of one at the expense of others) for each species and compares this to the absolute boundaries of size and economics for terrestrial vascular plants worldwide, thereby adhering to the foundational principles of plant-strategy theory.

We standardized species names via the R package *Taxonstand* (version 2.1)[73], using The Plant List (https://wfoplantlist.org/) as the backbone, to have a taxonomically-consistent dataset. Genome sizes were derived from the Plant DNA C-values database[74], which contains genome size data for 12,273 species. We cleaned the genome size data following Pyšek et al.[17], i.e., in case of multiple records for one species, we only included genome sizes estimated by flow cytometry with the intercalating dye propidium iodide; and preferred values that were (i) reported by authors from well-established laboratories, (ii) newly and repetitively estimated, and (iii) marked by authors of the Plant DNA C-values database as prime values. Here, the holoploid genome size (1C-values) refers to the total amount of DNA in the nucleus, whereas the monoploid genome size (1Cx-values) is calculated by taking into account the ploidy level and hence corresponds to the amount of DNA in one monoploid chromosome set, and thus allows comparisons to be made in DNA amounts that are independent of ploidy level[75]. Altogether, 1,612 species with available CSR scores had data on holoploid genome size. Among these, 993 species have known ploidy levels, which were used to calculate monoploid genome sizes.

### Economic uses

Data on economic uses were collated from the World Checklist of Useful Plant Species (WCUP; https://kew.iro.bl.uk/concern/datasets/7243d727-e28d-419d-a8f7-9ebef5b9e03e?)[76]. WCUP provides information on the economic use of 40,292 species. The economic uses are classified into 10 categories (e.g., animal food and environmental uses; see Supplementary Table 2 for details). Because it is likely that species are more widely cultivated if they have more economic uses, we included the number of economic use types in our analyses. If a species was missing from the WCUP checklist, its economic use was assigned a value of zero (i.e., no known economic use). Since the economic use

categories 'gene sources' and 'poisons' do not necessarily require cultivation of the species, and because these categories were shown to contribute little to plant naturalization success[11], we excluded these two categories from the assessment of species economic use.

### Native range size

Native range size was expressed as the number of TDWG level 3 regions where species have been recorded as native. This information was derived from multiple databases, i.e., Plants of the World Online (POWO, https://powo.science.kew.org)[77,78], the Global Compositae Database (GCD, https://www.compositae.org)[79], USDA GRIN-Global (GRIN, https://npgsweb.ars-grin.gov/gringlobal/search)[80], and the IUCN Red list (IUCN, https://www.iucnredlist.org)[81], in order of prioritization.

### Measures of naturalization and invasion success

Following previous studies[48,82], naturalized species are defined as alien species that can maintain self-sustaining populations without human intervention. Naturalization incidence (i.e., whether a species is naturalized or not) and extent (i.e. the number of regions that species are recorded in as naturalized, for the subset of species naturalized in at least one region) were extracted from the GloNAF database[83]. In our holoploid- and monoploid-focused dataset, 1193 and 792 species, respectively, were identified as naturalized.

To classify species as invasive, we followed the definition used in environmental policy—a subset of naturalized species that have been assessed as having negative impacts on the environment[84,85]. Although this definition differs from the one widely used in ecology that emphasizes the rapid spread of invasive species, several major databases have employed the impact-based definition. Therefore, this definition was used to gain a comprehensive and comparable compilation of invasion extent—the number of regions in which a naturalized species has been reported as invasive. Three global data sources: the CABI Invasive Species Compendium (https://www.cabi.org/isc)[86], the ISSG Global Invasive Species Database (https://www.iucngisd.org/gisd/)[87] and the invasive plant species database[88], were used for the compilation. In the holoploid- and monoploid-focused datasets, we identified 618 and 455 species, respectively, as invasive species.

### Phylogeny

Phylogenetic trees of species in the holoploid and monoploid dataset were obtained by pruning a megatree of vascular plants[89,90] to the species that were included in our datasets using the *V.PhyloMaker* (version 0.1.0) package[91]. Briefly, species missing from the backbone tree were added as polytomies to the middle of the branch of the species' genus or, if not available, the species' family (node = "build.nodes.1" and scenarios = 'S3'). We used R package *ggtree* (version 3.6.2)[92] for the visualization of phylogenetic tree and associated data.

### Data analysis

All data analyses were conducted in R (version 4.1)[93]. We fitted Bayesian structural equation models (SEMs) to test the direct and indirect links of multiple factors on the three measures of plant invasion success while accounting for phylogenetic relationships among species using the R package *brms* (version 2.19.0)[58]. Models that encapsulate the hypothesized relationships between the variables of interest (e.g., genome size → C-scores) were formulated using the function *bf* in the *brms* package. In these models, we included species as a random effect with a phylogenetic correlation structure (obtained via the function *vcv* in the R package *ape* (version 5.7.1)[94]) to account for phylogenetic autocorrelation[95]. In addition, we used flat priors[58] for the population-level (fixed) effects e.g., genome size → CSR scores. For the group-level effects such as intercept and slope variances of phylogenetic effects[95], Student's t-distributions with 3 degrees of freedom, a mean of 0 and a

scale of 2.5 were employed. We modelled binary response variables (e.g., genome size → naturalization incidence) using a Bernoulli distribution with a logit link function, and continuous variables (e.g., genome size → CSR scores) using a Gaussian distribution. These models were then aggregated into the SEM framework using the function *brm* in the *brms* package[96]. Note that even though paths included in our SEMs, e.g., direct and indirect paths (via economic uses) between native range size and plant invasion metrics (illustrated in Fig. 1), are not necessarily causal relationships, we used SEMs to rigorously test the ecological hypothesis-driven relationships outlined in our conceptual framework (Fig. 1).

Since CSR scores sum to 100% and thus are not independent, we considered each score in a separate model. Species' native range size, naturalization extent, and invasion extent were log-transformed and all continuous variables were standardized to mean = 0 and SD = 1. All models were run with four chains, 2000 iterations and a warmup of 1,000 iterations, and converged with R̂ values close to 1 (Gelman–Rubin diagnostic). To specifically examine the effect of each path in the SEMs, we calculated the one-tailed probability that the estimated coefficients were > 0 ($P_{est>0}$, for positive estimates) or <0 ($P_{est<0}$, for negative estimates).

Note that for SEMs of invasion extent, naturalization extent was included as an explanatory variable. All of the above analyses were conducted separately on the holoploid- and monoploid-focused dataset. Nevertheless, since the results of the two datasets were largely consistent with each other, we focused on the results of holoploid genome size in the main text and presented the results from the monoploid-focused dataset in the Supplementary material.

### Reporting summary
Further information on research design is available in the Nature Portfolio Reporting Summary linked to this article.

## Data availability
The databases that we used are all publicly available: The Plant List (https://wfoplantlist.org/); the Plant DNA C-values database (https://cvalues.science.kew.org/); World Checklist of Useful Plant Species (WCUP, https://kew.iro.bl.uk/concern/datasets/7243d727-e28d-419d-a8f7-9ebef5b9e03e?); Plants of the World Online (https://powo.science.kew.org); the Global Compositae Database (GCD, https://www.compositae.org); the USDA GRIN-Global (GRIN, https://npgsweb.ars-grin.gov/gringlobal/search); the IUCN Red list (IUCN, https://www.iucnredlist.org); Global Naturalized Alien Flora [GloNAF] https://doi.org/10.1002/ecy.2542); CABI Invasive Species Compendium (https://www.cabi.org/isc); ISSG Global Invasive Species Database (https://www.iucngisd.org/gisd/); Smith and Brown phylogenetic tree: https://github.com/FePhyFoFum/big_seed_plant_trees. The data (.RData file) that support the findings of this study are available on Github (https://github.com/kun-ecology/WorldPlantInvasion[97]) and are mirrored on Zenodo (https://doi.org/10.5281/zenodo.10113290[97]).

## Code availability
The scripts for reproducing the structural equation models and related visualizations are available on Github (https://github.com/kun-ecology/WorldPlantInvasion[97]) and are mirrored on Zenodo (https://doi.org/10.5281/zenodo.10113290[97]).

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

## Acknowledgements

KG was supported by the Shanghai Sailing Program (grant 22YF1411700) and the Natural Science Foundation of China (Grant 32301386). WYG and KG were supported by the Natural Science Foundation of China (Grant 32171588 to WYG) and the Shanghai Pujiang Program (grant 21PJ1402700 awarded to WYG). WYG was supported by the Innovation Program of Shanghai Municipal Education Commission (2023ZKZD36). NLK and MvK were supported by the German Research Foundation DFG (Grants 264740629 and 432253815 to MvK). PP and JP were supported by EXPRO grant no. 19-28807X (Czech Science Foundation) and with ML by long-term research development project RVO 67985939 (Czech Academy of Sciences). FE and BL were supported by the Austrian Science Foundation (Grant/Award Number: I 5825- B).

## Author contributions

KG, PP and WYG conceptualized the research; PP, MvK, NLK, ML, IJL, SP, WD, FE, BL, HK, JP, PW and WYG provided the data; KG analysed the data and drafted the manuscript; all authors contributed to revisions of the manuscript and gave final approval for publication.

## Competing interests

The authors declare no competing interests.

## Additional information

[1]Zhejiang Tiantong Forest Ecosystem National Observation and Research Station, School of Ecological and Environmental Sciences, East China Normal University, 200241 Shanghai, P. R. China. [2]Research Center for Global Change and Complex Ecosystems, School of Ecological and Environmental Sciences, East China Normal University, 200241 Shanghai, P. R. China. [3]Czech Academy of Sciences, Institute of Botany, Department of Invasion Ecology, Průhonice CZ-25243, Czech Republic. [4]Department of Ecology, Faculty of Science, Charles University, Viničná 7, Prague CZ-12844, Czech Republic. [5]Ecology, Department of Biology, University of Konstanz, Universitätsstrasse 10, D-78457 Konstanz, Germany. [6]Zhejiang Provincial Key Laboratory of Plant Evolutionary Ecology and Conservation, Taizhou University, Taizhou 318000, P. R. China. [7]Czech Academy of Sciences, Institute of Botany, Department of Evolutionary Plant Biology, Průhonice CZ-25243, Czech Republic. [8]Department of Botany, Faculty of Science, University of South Bohemia, Branišovská 1760, České Budějovice CZ-370 05, Czech Republic. [9]Royal Botanic Gardens, Kew, Richmond, Surrey TW9 3AB, UK. [10]Department of Agricultural and Environmental Sciences (DiSAA), University of Milan, Via G. Celoria 2, I-20133 Milan, Italy. [11]Department of Biosciences, Durham University, Durham, UK. [12]Department of Evolution, Ecology and Behaviour, Institute of Infection, Veterinary and Ecological Sciences, University of Liverpool, Liverpool, UK. [13]Division of BioInvasions, Global Change & Macroecology, Department of Botany and Biodiversity Research, University of Vienna, Vienna, Austria. [14]Biodiversity, Macroecology & Biogeography, University of Goettingen, Göttingen, Germany. [15]Centre of Biodiversity and Sustainable Land Use (CBL), University of Goettingen, Göttingen, Germany. [16]Campus-Institute Data Science, Göttingen, Germany. [17]Shanghai Key Lab for Urban Ecological Processes and Eco-Restoration, School of Ecological and Environmental Sciences, East China Normal University, 200241 Shanghai, P. R. China. ✉e-mail: wyguo@des.ecnu.edu.cn

