## [Peer Review File · Nature Communications]

Reviewers' Comments:

Reviewer #1:

Remarks to the Author:

General Comments:

This is an interesting and important paper. I was particularly impressed by the ability of the authors to consider a large number of simultaneous process and still summarized the general findings in a clear and interpreted fashion.

I have been asked to focus on the SEM in the paper, so will not comment on the open question of whether the C-S-R scheme is the ideal one for dealing with plant trait syndromes.

Review Comments Related to the SEM analyses:

1. SEM analyses range from those with scant explicit scientific justification to those that are more advanced and provide a more complete documentation of underlying supportive knowledge. I was pleased to see Table 1 in this paper providing summaries of our pre-existing knowledge related to the model in Figure 1, along with a thorough discussion of the key knowledge in the Introduction.
2. The authors used BRMS to implement Bayesian SEM.
3. The authors ran separate SEMs for the CSR elements to avoid non-independence.
4. To account for phylogenetic autocorrelation the authors included species as a random effect with a phylogenetic correlation structure.
5. I note that the authors model binary responses using a logit link structure.
6. I note that the Figures presents standardized coefficients.
7. The supplementary file presents additional details regarding results, though no further description of the SEM procedures.
8. A file containing the data and code are referenced in the manuscript, however, reviewers are not provided access at this time.
9. The only reference to SEM methods and assumptions is the statement, "... we used Bayesian structural equation models (SEMs) to assess how the identified factors affect three different metrics of invasion success (as in Razanajatovo et al. 2016; Gioria et al. 2021)"
10. Regarding the Razanaiatovo reference, the analysis used ML methods, specifically those implemented in the lavaan package, with methodological references given. The Gioria reference also relied on the lavaan package and provided amply sources to literature relating to the key features of the SEM analyses.
11. Since papers are supposed to provide sufficient information to permit others to competently follow up on the work presented, I must point out that no references for Bayesian SEM are presented, which can be easily corrected I think by citing:

Grace, J. B., Schoolmaster Jr, D. R., Guntenspergen, G. R., Little, A. M., Mitchell, B. R., Miller, K. M., & Schweiger, E. W. (2012). Guidelines for a graph-theoretic implementation of structural equation modeling. *Ecosphere*, 3(8), 1-44.

This paper provides a detailed description of the SE modeling process featuring a Bayesian modeling specification. The Bayesian specification used in the manuscript appears to be somewhat more complex, but I cannot judge exactly what additional references if any would be helpful for the reader interested in following up on the work presented.

12. Another point I will mention relates to the business of presenting standardized coefficients involving binary responses, which are included in this paper. I am not requesting that the authors redo their standardizations. However, this seems a good time for them to be aware of the inherent problems and the available solutions (that have taken some years of effort to craft). I would suggest reading:

Grace, J. B., Johnson, D. J., Lefcheck, J. S., & Byrnes, J. E. (2018). Quantifying relative importance: computing standardized effects in models with binary outcomes. *Ecosphere*, 9(6), e02283.

A general method of standardization is also provided in the Grace et al. 2012 paper and code is provided for its implementation in the Supplementary Material for that paper. The Supplementary

Material for the 2018 paper provides a wider variety of approaches.

Minor Issues:

1. I notice a typo at the end of line 119.
2. Line 121 – should be “strategies”.

Reviewer #2:

Remarks to the Author:

In this paper Guo et al. describe the importance of 4 very different variables to explain biological invasions. The factors are genome size, Grime’s adaptive strategies, native range size and economic use. Given how different the variables studied are, it is not surprising that the results are very novel. I think that this paper is exceptionally well conducted and that this is a solid contribution to the field of invasion biology. Below I have some concerns that should be addressed before the paper is published.

Title: I think that the title is a bit misleading. These is a study on a very peculiar sub set of characteristic of the plants and took me a while that this was not a comprehensive analysis of more plant specific characteristics that were also used before (e.g. seed size, growth rates, LAI) and are more specific in a way than Grime’s adaptive strategies which also of course encompasses all these traits but by combining them sometimes. So I think that the title should make clear that there were 4 factors analyzed. Also, this may be because I not a native English speaker, but the word "Exacerbate" was a bit confusing for me, why not using “promote”?

After reading the abstract for the 1st time it was not clear what where the main results. Perhaps the title confused me. But why you decided to study these 4 very different variables perhaps was why it was more confusing (e.g. genome size and economic use is not something that I would have think have some relationship to explain invasion success). A bit more info on that would be great.

It would be nice to see more references to the new IPBES assessment on invasive species in the references, since it really related to some aspects of this research (the magnitude of the problem, the drivers, etc.)

All in all, I consider that this is a very interesting and thought-provoking paper. Also, it is the product of decades of research and amazing data. I think that some aspects need to be improved (e.g. title and abstract) and I am still a bit puzzled about its structure (i.e. why did you compare these 4 variables and not others), but I think thing this is a very important contribution that may open new lines of research.

Reviewer #3:

Remarks to the Author:

Guo et al., Plants used by humans have characteristics that exacerbate invasions worldwide. Nature Communications.

This is a useful exploration of the potential mechanisms by which genome size may influence invasion. It is generally well written and I had no trouble understanding the authors’ main points. The manuscript complements other work by this group exploring how invasion is made up of multiple ecological processes, and plant traits influence these in different ways.

A key strength of this study is considering the factors influencing naturalization separate from the factors influencing invasiveness. However, it is misleading to claim, as this manuscript does, that it is a comprehensive study across all the stages and processes of invasion. As they describe on page 7, the stages include human-mediated introduction, establishment, then naturalization, then invasion. This work only addresses the last two stages. This is one example of several ways in

which this paper, which is a solid contribution on its own merits, overstates its generality and scope (e.g. lines 260-1).

If, as suggested by the title, the main goal of the work is to explore how human uses of plants influence naturalization and invasion, then the paper misses its mark. For that goal, the authors might instead explore the influence of each of the many types of uses that are included in the WCUP database. However, it is probably easier to change the title and framing. Isn't the focus more on the many ways that genome size may influence invasion?

The issue of polyploidy is interesting and important. In fact, the final sentences of the paper argue that the effect of large-genome plants on life history is most likely driven by polyploidy. Yet in the manuscript, the holoploid dataset is emphasized over the monoploid dataset, with the justification that results are similar. After reading that, I was later surprised by lines 217-18, which describe seemingly important differences, and comparing Figure 3 and Figure S4 in detail, the differences are substantial for relationships with genome size. It seems like this study could do more to explore the dual influences of polyploidy and monoploid genome size. I wondered whether the SEM approach could be harnessed to consider both polyploidy and the monoploid genome size in the same model, and thereby disentangle some of the multiple ways that genome size might influence invasion.

The power of taking an SEM approach is potentially that it allows the testing of complex direct and indirect paths of causality. This was my expectation of this manuscript, and so I was somewhat disappointed that it did not offer deeper insights into mechanisms and explanations. Instead, the manuscript left me with a number of questions about causality. For example, does geographic range size actually "influence" success (as stated in lines 95-6), or does it simply predict success because of some underlying shared driver? Lines 138-149 implies the latter with its discussion of "positive correlation"...but then the SEM models all have direct arrows from range size to naturalization, which is kind of confusing. Similarly, for Table 1#8 ["Plants with wide native ranges are more frequently used by humans"], is there an implication of some kind of mechanism here? This is important in order to understand whether the argument is about propagule pressure via human introduction, or some other driver that increases range size, thereby increasing the probability that the species has multiple human uses as well as (but independent of) increasing the probability that the species will establish in a new range.

It must have been challenging to decide how to quantify "economic usefulness" from the data available. I went to the WCUP to explore the database and noticed a couple of things that made me question the results. Many of the species there have as their primary/only 'economic use' that they are phylogenetically related to a crop or other economically useful species (= the "GS" designation). Just because the species has a close relative that is a crop does not imply anything about the species' ecology or its own relationship to human activities. It seems to me that this study should not include species that only have a "GS" designation as having an economic use. Similarly but less importantly, the "PO" designation in WCUP seems to include both species with poisons that are actively used by people, but also species that are just poisonous and not necessarily used to extract poisons. I'm skeptical of that category as well. Perhaps these details would make no difference to the analysis, since "usefulness" is quantitative (number of categories) rather than binary. However, I think it is important to try running the analyses with at least the GS category (or possibly both GS and PO) removed.

It is unclear how the two measures of naturalization: naturalized incidence vs. extent, are thought to map separately onto the stages of invasion—this is related to my earlier comment wishing for a clearer exploration of mechanism and causality. In fact, I am not sure that it is necessary/desirable to include both naturalization incidence and extent as independent sets of analyses; Figures 3 and S4 suggest that these two metrics show very similar patterns and perhaps extent is just a more sensitive metric for revealing the effects of genome size.

It is quite remarkable to me that of the 1,612 plants for which holoploid genome sizes are available, a full 618 of them are known invasive species. This underscores that these 1,612 are definitely not a random subset of plant diversity, and I wonder how that affects the conclusions we can draw. I recognize that this is the dataset we have to work with, so this isn't a criticism of the

project, it just raises questions for me.

Line 347 (Methods): It is very important to add some details here about how the C, S, & R scores are generated. I would rather not have to go read Pierce et al. 2017 in order to discover that these scores are based on just two variables: leaf area and leaf dry matter content (and SLA, which is a linear combination of those two variables).

Line 62, 263 & elsewhere: I find the term "hierarchical network" confusing—it does not seem appropriate to call this phenomenon either a network or hierarchical.

Line 89 etc.: I suggest the term "economically useful" over "economically used".

Line 143: I suggest "species characterized by traits that promote colonization"

I liked Figure 2 and found that it included a lot of information efficiently. I particularly liked how CSR strategies were represented as colors dividing a single bar.

In summary, this study takes an interesting approach to some interesting questions about the drivers of invasion. I hope my suggestions may help improve the manuscript and/or perhaps suggest some future follow-on studies.

Sincerely,
Ingrid M. Parker

Reviewer #1

General Comments:

1. This is an interesting and important paper. I was particularly impressed by the ability of the authors to consider a large number of simultaneous process and still summarized the general findings in a clear and interpreted fashion.

We thank the reviewer for the positive feedback.

2. I have been asked to focus on the SEM in the paper, so will not comment on the open question of whether the C-S-R scheme is the ideal one for dealing with plant trait syndromes.

We thank the reviewer for the thorough evaluation and insightful comments on the SEM analyses.

3. Review Comments Related to the SEM analyses:

1) SEM analyses range from those with scant explicit scientific justification to those that are more advanced and provide a more complete documentation of underlying supportive knowledge. I was pleased to see Table 1 in this paper providing summaries of our pre-existing knowledge related to the model in Figure 1, along with a thorough discussion of the key knowledge in the Introduction.

2) The authors used BRMS to implement Bayesian SEM.

3) The authors ran separate SEMs for the CSR elements to avoid non-independence.

4) To account for phylogenetic autocorrelation the authors included species as a random effect with a phylogenetic correlation structure.

5) I note that the authors model binary responses using a logit link structure.

6) I note that the Figures presents standardized coefficients.

We thank the reviewer for this summary.

7) The supplementary file presents additional details regarding results, though no further description of the SEM procedures.

We thoroughly revised the text and added more details to the description of the SEM procedures in lines 347–361, which now reads: "Models that encapsulate the hypothesized relationships between the variables of interest (e.g., genome size → C-scores) were formulated using the function *bf* in the *brms* package. In these models, we included species as a random effect with a phylogenetic correlation structure (obtained via the function *vcv* in the R package *ape*⁹²) to account for phylogenetic autocorrelation⁹³. In addition, we used flat priors⁵⁶ for the population-level (fixed) effects e.g., genome size → CSR scores. For the group-level effects such as intercept and slope variances of phylogenetic effects⁹³, Student' s t-distributions with 3 degrees of freedom , a mean of 0 and a scale of 2.5 were employed. We modelled binary response variables (e.g., genome

size → naturalization incidence) using a Bernoulli distribution with a logit link function, and continuous variables (e.g., genome size → CSR scores) using a Gaussian distribution. These models were then aggregated into the SEM framework using the function *brm* in the *brms* package⁹⁴. Note that even though paths included in our SEMs e.g., direct and indirect paths (via economic uses) between native range size and plant invasion metrics (illustrated in Fig. 1) are not necessarily causal relationships, we used SEMs to rigorously test the ecological hypothesis-driven relationships outlined in our conceptual framework (Fig. 1)."

8) A file containing the data and code are referenced in the manuscript, however, reviewers are not provided access at this time.

We are sorry for the inconvenience. The updated code and data have now been deposited on Github (<https://github.com/kun-ecology/WorldPlantInvasion>) and are mirrored on Zenodo (<https://doi.org/10.5281/zenodo.10113290>).

9) The only reference to SEM methods and assumptions is the statement, "... we used Bayesian structural equation models (SEMs) to assess how the identified factors affect three different metrics of invasion success (as in Razanajatovo et al. 2016; Gioria et al. 2021)"

10) Regarding the Razanaiatovo reference, the analysis used ML methods, specifically those implemented in the lavaan package, with methodological references given. The Gioria reference also relied on the lavaan package and provided ample sources to literature relating to the key features of the SEM analyses.

11) Since papers are supposed to provide sufficient information to permit others to competently follow up on the work presented, I must point out that no references for Bayesian SEM are presented, which can be easily corrected I think by citing: Grace, J. B., Schoolmaster Jr, D. R., Guntenspergen, G. R., Little, A. M., Mitchell, B. R., Miller, K. M., & Schweiger, E. W. (2012). Guidelines for a graph-theoretic implementation of structural equation modeling. *Ecosphere*, 3(8), 1-44.

We thank the reviewer for pointing this out. Although Razanajatovo *et al.* (2016) and Gioria *et al.* (2021) used different packages and structures to run SEMs, we included their papers here also because they used multiple metrics to explain invasion success. We have now modified the sentence to clarify this and further cited Grace et al. (2012) and Bürkner (2017) as references for Bayesian SEM. Please see lines 133–135: " Specifically, we used Bayesian structural equation models (SEMs)^{56,57} to assess how these factors affect multiple metrics of invasion success^{17,58,59}".

12) This paper provides a detailed description of the SE modeling process featuring a Bayesian modeling specification. The Bayesian specification used in the manuscript appears to be somewhat more complex, but I cannot judge exactly what additional references if any would be helpful for the reader interested in following up on the work presented.

To help the reviewer and future readers to better understand the rather complicated analysis, we have included more details on SEM model specification in lines 347–358: "Models that encapsulate the hypothesized relationships between the variables of interest (e.g., genome size → C-scores) were formulated using the function *bf* in the *brms* package. In these models, we included species as a random effect with a phylogenetic correlation structure (obtained via the function *vcv* in the R package *ape*⁹²) to account for phylogenetic autocorrelation⁹³. In addition, we used flat priors⁵⁶ for the population-level (fixed) effects e.g., genome size → CSR scores. For the group-level effects such as intercept and slope variances of phylogenetic effects⁹³, Student's t-distributions with 3 degrees of freedom, a mean of 0 and a scale of 2.5 were employed. We modelled binary response variables (e.g., genome size → naturalization incidence) using a Bernoulli distribution with a logit link function, and continuous variables (e.g., genome size → CSR scores) using a Gaussian distribution. These models were then aggregated into the SEM framework using the function *brm* in the *brms* package⁹⁴."

13) Another point I will mention relates to the business of presenting standardized coefficients involving binary responses, which are included in this paper. I am not requesting that the authors redo their standardizations. However, this seems a good time for them to be aware of the inherent problems and the available solutions (that have taken some years of effort to craft). I would suggest reading: Grace, J. B., Johnson, D. J., Lefcheck, J. S., & Byrnes, J. E. (2018). Quantifying relative importance: computing standardized effects in models with binary outcomes. *Ecosphere*, 9(6), e02283. A general method of standardization is also provided in the Grace et al. 2012 paper and code is provided for its implementation in the Supplementary Material for that paper. The Supplementary Material for the 2018 paper provides a wider variety of approaches.

We thank the reviewer for the kind recommendation. We acknowledge the complexities and challenges in presenting standardized coefficients involving binary responses and the great efforts many scientists have devoted to this. The suggested references to Grace et al. (2012, 2018) are highly appreciated.

Minor Issues:

4. I notice a typo at the end of line 119.

Revised.

5. Line 121 – should be “strategies”.

Revised.

Reviewer #2

1. In this paper Guo et al. describe the importance of 4 very different variables to explain biological invasions. The factors are genome size, Grime's adaptive strategies, native range size and economic use. Given how different the variables studied are, it is not surprising that the results are very novel. I think that this paper is exceptionally well conducted and that this is a solid contribution to the field of invasion biology. Below I have some concerns that should be addressed before the paper is published.

We thank the reviewer for the encouraging comments on our work.

Title: I think that the title is a bit misleading. This is a study on a very peculiar sub set of characteristic of the plants and took me a while that this was not a comprehensive analysis of more plant specific characteristics that were also used before (e.g. seed size, growth rates, LAI) and are more specific in a way than Grime's adaptive strategies which also of course encompasses all these traits but by combining them sometimes. So I think that the title should make clear that there were 4 factors analyzed. Also, this may be because I am not a native English speaker, but the word "Exacerbate" was a bit confusing for me, why not using "promote"?

We appreciate the suggestions on the title, which did not highlight the multifactor analyses done in our study. However, we also felt that there would be too much detail in the title if we explicitly stated that four factors were analyzed. In addition, we note that Grime's adaptive strategies encompass multiple traits that represent the principle functional space of plants worldwide i.e., the trade-off between resource economics and size, with which further traits such as seed mass are correlated (see the multivariate analyses in Díaz et al. (2016) and Pierce et al. (2017) for details; both papers are cited in the present study). Given the above, we have revised the title to: "Plant invasion success worldwide: the interplay between plant characteristics and economic uses".

2. After reading the abstract for the 1st time it was not clear what where the main results. Perhaps the title confused me. But why you decided to study these 4 very different variables perhaps was why it was more confusing (e.g. genome size and economic use is not something that I would have think have some relationship to explain invasion success). A bit more info on that would be great.

The rationale for choosing the four factors has been thoroughly justified in the Introduction. However, we now realize that due to the word limit, the rationale might not have been so clear from the abstract. We have therefore revised the abstract to explain that these variables have been shown to relate to invasion success but have not previously been studied jointly. Please see lines 40-43: " However, while previous studies often examined a limited number of factors or focused on a specific invasion stage (e.g., naturalization) for specific regions, a multi-factor and multi-stage analysis at the global scale is lacking."

3. It would be nice to see more references to the new IPBES assessment on invasive species in the references, since it really related to some aspects of this research (the magnitude of the problem, the drivers, etc.)

Thanks for the advice. We have now cited the latest IPBES assessment (IPBES 2023) (see reference 7) in the following places of the main text where relevant:

- Lines 59–60: "Invasive species also affect human well-being and cause substantial economic losses globally^{6–8}".
- Lines 60–62: " With the world facing growing anthropogenic pressures and becoming increasingly interconnected, projections indicate a surge in the number of alien species in future decades^{7,9,10}".

4. All in all, I consider that this is a very interesting and thought-provoking paper. Also, it is the product of decades of research and amazing data. I think that some aspects need to be improved (e.g. title and abstract) and I am still a bit puzzled about its structure (i.e. why did you compare these 4 variables and not others), but I think thing this is a very important contribution that may open new lines of research.

We thank the reviewer again for the encouraging and insightful comments on our work. We have revised the title and provided more details on why we selected the four variables as stated above, and we hope the reviewer will find these revisions satisfactory.

Reviewer #3

1. This is a useful exploration of the potential mechanisms by which genome size may influence invasion. It is generally well written and I had no trouble understanding the authors' main points. The manuscript complements other work by this group exploring how invasion is made up of multiple ecological processes, and plant traits influence these in different ways.

A key strength of this study is considering the factors influencing naturalization separate from the factors influencing invasiveness. However, it is misleading to claim, as this manuscript does, that it is a comprehensive study across all the stages and processes of invasion. As they describe on page 7, the stages include human-mediated introduction, establishment, then naturalization, then invasion. This work only addresses the last two stages. This is one example of several ways in which this paper, which is a solid contribution on its own merits, overstates its generality and scope (e.g. lines 260-1).

We appreciate the reviewer's positive feedback on our manuscript. Indeed, we lack data on the "introduction" stage", as discussed in lines 247–249: "Specifically, we lacked data that directly capture introduction effort and casual occurrences, meaning that possible filter effects of the early invasion stages are missing". We therefore now more carefully refer to the multi-stage process of plant invasion. For example, we have revised the text in lines 192–194 to read: "By compiling a global dataset and integrating data on factors that have been shown to be related to invasion success, we explored how species characteristics and their usefulness for humans affects plant naturalization and invasion at the global scale."

2. If, as suggested by the title, the main goal of the work is to explore how human uses of plants influence naturalization and invasion, then the paper misses its mark. For that goal, the authors might instead explore the influence of each of the many types of uses that are included in the WCUP database. However, it is probably easier to change the title and framing. Isn't the focus more on the many ways that genome size may influence invasion?

We thank the reviewer for this insightful comment. We agree that the title might not have clearly conveyed the main goal of our study i.e., the multiple-factor perspective on plant naturalization and invasion worldwide. Based on this comment and the suggestion by Reviewer 2, we have revised the title to: "Plant invasion success worldwide: the interplay between plant characteristics and economic uses".

3. The issue of polyploidy is interesting and important. In fact, the final sentences of the paper argue that the effect of large-genome plants on life history is most likely driven by polyploidy. Yet in the manuscript, the holoploid dataset is emphasized over the monoploid dataset, with the justification that results are similar. After reading that, I was later surprised by lines 217-18, which describe seemingly important differences, and comparing Figure 3 and Figure S4 in detail, the differences are substantial for relationships with genome size. It seems like this study could do more to explore the dual influences of polyploidy and monoploid genome size. I wondered whether the SEM approach could be harnessed to consider both polyploidy and the monoploid genome size in the same model, and thereby disentangle some of the multiple ways that genome size might influence invasion.

We concur that both genome size and ploidy levels play important roles in plant naturalization and invasion. This perspective is corroborated by the recent study by Pyšek *et al.* (2023), who demonstrated the significant roles of these two factors in affecting plant invasions using the Plant DNA C-values database (<https://cvalues.science.kew.org>). Through analyses with 11,049 taxa, their results showed that a small genome (both holoploid and monoploid) and polyploidy favour naturalization but limit invasive spread, resulting in a hump-shaped relationship between genome size and plant invasion. In contrast to their study, which established a comprehensive understanding of the direct pathways of the relationships between genomic characteristics and invasion success, our study integrates several species characteristics that potentially are related to genome size, to gain a multifactor view of plant invasion. Indeed, our results revealed several indirect pathways of how genome size is related to naturalization and invasion success. Therefore, we are confident that without introducing another genomic variable and the many pathways in which it may be linked to other variables to our analytical framework, our study still provides insights regarding the nuanced interplay between genome size and plant invasion.

4. The power of taking an SEM approach is potentially that it allows the testing of complex direct and indirect paths of causality. This was my expectation of this manuscript, and so I was somewhat disappointed that it did not offer deeper insights into mechanisms and explanations. Instead, the manuscript left me with a number of questions about causality. For example, does geographic range size actually "influence" success (as stated in

lines 95-6), or does it simply predict success because of some underlying shared driver? Lines 138-149 implies the latter with its discussion of “positive correlation”...but then the SEM models all have direct arrows from range size to naturalization, which is kind of confusing. Similarly, for Table 1#8 [“Plants with wide native ranges are more frequently used by humans”], is there an implication of some kind of mechanism here? This is important in order to understand whether the argument is about propagule pressure via human introduction, or some other driver that increases range size, thereby increasing the probability that the species has multiple human uses as well as (but independent of) increasing the probability that the species will establish in a new range.

#SEMs are used to analyse direct and indirect causal relationships among the variables included in the analysis. This, however, does not necessarily exclude the possibility that observed direct causal relationships are in reality also driven by indirect relationships that were not included in the SEM. In the case of native range size, there are different traits that may determine it, and there are multiple reasons why it might positively affect naturalization and invasion success. In our SEM, we tried to partly unravel them, as we allowed native range size to be determined by genome size and the CSR scores, and we allowed native range size to affect naturalization and invasion success via its effects on economic use. Nevertheless, the remaining direct effect of native range size could still be mediated by other variables not included in the SEM. We now explain this more carefully in the manuscript, please see lines 358–361: “Note that even though paths included in our SEMs e.g., direct and indirect paths (via economic uses) between native range size and plant invasion metrics (illustrated in Fig. 1) are not necessarily causal relationships, we used SEMs to rigorously test the ecological hypothesis-driven relationships outlined in our conceptual framework (Fig. 1).”

5. It must have been challenging to decide how to quantify “economic usefulness” from the data available. I went to the WCUP to explore the database and noticed a couple of things that made me question the results. Many of the species there have as their primary/only ‘economic use’ that they are phylogenetically related to a crop or other economically useful species (= the “GS” designation). Just because the species has a close relative that is a crop does not imply anything about the species’ ecology or its own relationship to human activities. It seems to me that this study should not include species that only have a “GS” designation as having an economic use. Similarly but less importantly, the “PO” designation in WCUP seems to include both species with poisons that are actively used by people, but also species that are just poisonous and not necessarily used to extract poisons. I’m skeptical of that category as well. Perhaps these details would make no difference to the analysis, since “usefulness” is quantitative (number of categories) rather than binary. However, I think it is important to try running the analyses with at least the GS category (or possibly both GS and PO) removed.

We thank the reviewer for pointing this out. It is true that both the category gene sources (GS) and poisons (PO) are not significantly related to plant naturalization, as shown by van Kleunen *et al.* (2020; see Fig. 2 in their study for details). We followed the reviewer’s suggestion and reran the analyses with both GS and PO removed. The results remained largely the same (compare the figure below with the previous Fig. 3). The only small qualitative change is that the new results show that economic use is positively related to invasion extent,

further highlighting the important role of economic use in invasion success. Therefore, we updated the methods and have now included the new SEM results in the revised manuscript. Please, see lines 308–311 for details: "Since the economic use categories 'gene sources' and 'poisons' do not necessarily require cultivation of the species, and because these categories were shown to contribute little to plant naturalization success¹¹, we excluded these two categories from the assessment of species economic use."

Fig. 3 Structural equation models linking plant holoploid genome size, CSR-strategy scores, native range size, and economic use to (a-c) naturalization incidence (NatInd i.e., whether the species is listed as naturalized in the GloNAF database; n = 1,612), (d-f) naturalization extent (NatExt i.e., the number of regions a species has been recorded in as naturalized; n = 1,193), and (g-i) invasion extent (InvExt i.e., the number of regions a species has been recorded in as invasive; n = 618). Solid red lines, solid blue lines and dashed grey lines, respectively, indicate negative, positive and non-significant (95% credible interval includes zero) relationships. Numbers beside the arrows are standardized coefficients, which are only shown for significant paths. See Fig. S1-3 for details of each path.

6. It is unclear how the two measures of naturalization: naturalized incidence vs. extent, are thought to map separately onto the stages of invasion—this is related to my earlier comment wishing for a clearer exploration of mechanism and causality. In fact, I am not sure that it is necessary/desirable to include both naturalization incidence and extent as independent sets of analyses; Figures 3 and S4 suggest that these two metrics show very similar patterns and perhaps extent is just a more sensitive metric for revealing the effects of genome size.

We use both naturalization incidence and extent since they capture different facets of naturalization—essentially addressing whether a species can naturalize and, if it naturalizes, how widely it has naturalized. In other words, naturalization extent is only analysed for species that have become naturalized in at least one region, and this analysis thus provides more insight into what drives different aspects of plant naturalization. We now explain this more clearly in lines 136-137: "(ii) naturalization extent (in how many regions an alien species has naturalized, provided that it has naturalized in at least one region)"

Our approach also aligns with methodologies adopted in several previous studies (Gioria *et al.* 2021; Guo *et al.* 2019, 2018; Pyšek *et al.* 2023; Razanajatovo *et al.* 2016). Our SEM results on both metrics remain largely the same (Fig 3a-c vs. Fig.3 d-f), but notable differences are also observed. For instance, R-scores showed no significant relationship with naturalization incidence, but a positive relationship was found with naturalization extent. Such differences underscore the importance of including both metrics as they yield a more comprehensive understanding of plant naturalization.

7. It is quite remarkable to me that of the 1,612 plants for which holoploid genome sizes are available, a full 618 of them are known invasive species. This underscores that these 1,612 are definitely not a random subset of plant diversity, and I wonder how that affects the conclusions we can draw. I recognize that this is the dataset we have to work with, so this isn't a criticism of the project, it just raises questions for me.

#The notable presence of invasive species within the dataset of species with known holoploid genome sizes does indicate that the latter constitutes a non-random subset of plant diversity. The most likely explanation is that alien and particularly invasive plants attract more research attention. As there has been a long-lasting

interest in the genetics of invasive species (since the influential book by Baker and Stebbins (1965) on *The Genetics of Colonizing Species*), this research has as a 'by-product' resulted in an overproportional amount of genome size data for invasive plants. While of course we would ideally have genome size data for the entire global flora, we do not see how the biased availability of such data would affect the actual relationship of genome size with naturalization and invasion success.

8. Line 347 (Methods): It is very important to add some details here about how the C, S, & R scores are generated. I would rather not have to go read Pierce et al. 2017 in order to discover that these scores are based on just two variables: leaf area and leaf dry matter content (and SLA, which is a linear combination of those two variables).

As suggested, we added more details on the generation of the C, S and R scores to the methods. Please, see lines 268–284: "CSR scores were quantified based on three traits demonstrated to strongly represent the principle functional space of plants: leaf area (LA; representing the plant/organ size spectrum), specific leaf area (SLA; high values representing 'acquisitive' plant resource economics), and leaf dry matter content (LDMC; high values representing 'conservative' economics)²⁹. While only three traits suffice for CSR calculation, they also exhibit significant statistical correlations with a more extensive range of plant characteristics, encompassing whole plant traits (canopy height, lateral spread), leaf traits (leaf nitrogen and carbon concentrations), and reproductive traits (seed mass, seed volume, seed variance, total mass of seeds, flowering period and flowering start) in the world flora (see Pierce et al.²⁹ for a multivariate analysis of these relationships).

Data for these three traits were collated from multiple sources^{32,68–72}. In instances where multiple trait values were available for a species, we used the mean values for the CSR calculation. The 'StrateFy' CSR classification tool of Pierce et al.²⁹ employed here does not simply use each trait to directly represent each axis. Instead, it determines the trade-off between traits (i.e., increased values of one at the expense of others) for each species and compares this to the absolute boundaries of size and economics for terrestrial vascular plants worldwide, thereby adhering to the foundational principles of plant-strategy theory."

9. Line 62, 263 & elsewhere: I find the term "hierarchical network" confusing—it does not seem appropriate to call this phenomenon either a network or hierarchical.

We see the reviewer's point, and we now removed the term.

10. Line 89 etc.: I suggest the term "economically useful" over "economically used".

Revised.

11. Line 143: I suggest "species characterized by traits that promote colonization"

Revised.

12. I liked Figure 2 and found that it included a lot of information efficiently. I particularly liked how CSR strategies were represented as colors dividing a single bar.

We thank the reviewer for this compliment.

13. In summary, this study takes an interesting approach to some interesting questions about the drivers of invasion. I hope my suggestions may help improve the manuscript and/or perhaps suggest some future follow-on studies.

We thank the reviewer for the constructive feedback and encouraging words.

Papers cited:

- Gioria, M., Carta, A., Baskin, C.C., Dawson, W., Essl, F., Kreft, H., *et al.* (2021). Persistent soil seed banks promote naturalisation and invasiveness in flowering plants. *Ecol. Lett.*, 24, 1655–1667.
- Guo, K., Pyšek, P., Chytrý, M., Divíšek, J., Lososová, Z., van Kleunen, M., *et al.* (2022). Ruderals naturalize, competitors invade: Varying roles of plant adaptive strategies along the invasion continuum. *Funct. Ecol.*, 36, 2469–2479.
- Guo, W.-Y., Van Kleunen, M., Winter, M., Weigelt, P., Stein, A., Pierce, S., *et al.* (2018). The role of adaptive strategies in plant naturalization. *Ecol. Lett.*, 21, 1380–1389.
- Guo, W.-Y., van Kleunen, M., Pierce, S., Dawson, W., Essl, F., Kreft, H., *et al.* (2019). Domestic gardens play a dominant role in selecting alien species with adaptive strategies that facilitate naturalization. *Glob. Ecol. Biogeogr.*, 28, 628–639.
- IPBES. (2023). Summary for policymakers of the thematic assessment report on invasive alien species and their control of the intergovernmental science-policy platform on biodiversity and ecosystem services. In: (eds. Roy, H.E., Pauchard, A., Stoett, P., Renard Truong, T., Bacher, S., Galil, B.S., *et al.*). IPBES Secretariat, Bonn, Germany.
- van Kleunen, M., Xu, X., Yang, Q., Maurel, N., Zhang, Z., Dawson, W., *et al.* (2020). Economic use of plants is key to their naturalization success. *Nat. Commun.*, 11, 1–12.
- Pyšek, P., Lučanová, M., Dawson, W., Essl, F., Kreft, H., Leitch, I.J., *et al.* (2023). Small genome size and variation in ploidy levels support the naturalization of vascular plants but constrain their invasive spread. *New Phytol.*, 239, 2389–2403.
- Razanajatovo, M., Maurel, N., Dawson, W., Essl, F., Kreft, H., Pergl, J., *et al.* (2016). Plants capable of selfing are more likely to become naturalized. *Nat. Commun.* 2016 71, 7, 1–9.

Reviewers' Comments:

Reviewer #1:

Remarks to the Author:

The reviewers offered a suite of suggestions for improvement, which I believe the authors have satisfactorily addressed. I find the revised manuscript to be well crafted and interesting.

Reviewer #2:

Remarks to the Author:

I am very happy to see that the authors have addressed all of my main comments. This version is notably improved. Thanks so much for all of your work on this.

Reviewer #3:

Remarks to the Author:

I reviewed the Response to Reviewers and the revised manuscript, and I appreciate the changes made by the authors to address many of my suggestions. I am disappointed that the SEM approach was not applied to explore the dual roles of monoploid genome size and polyploidy. However, I recognize that that would have required a substantial amount of extra modeling, and presumably the authors did not want to do it.

Many of the responses to my review pointed to other related work by the authors. That's great, and their responses underscore the deep knowledge and insights that this team of authors have in the area of invasion ecology at the global scale. My additional suggestion would be to look through the manuscript again and make sure that the revised paper takes advantage of this previous work and points other readers (not just me as a reviewer) to those resources, findings, and insights. I think the paper could be slightly improved by drawing on these insights more fully.

Reviewer #1

The reviewers offered a suite of suggestions for improvement, which I believe the authors have satisfactorily addressed. I find the revised manuscript to be well crafted and interesting.

We thank the reviewer for the constructive feedback and encouraging words.

Reviewer #2

I am very happy to see that the authors have addressed all of my main comments. This version is notably improved. Thanks so much for all of your work on this.

We thank you for your insightful comments and encouraging words.

Reviewer #3

1. I reviewed the Response to Reviewers and the revised manuscript, and I appreciate the changes made by the authors to address many of my suggestions. I am disappointed that the SEM approach was not applied to explore the dual roles of monoploid genome size and polyploidy. However, I recognize that that would have required a substantial amount of extra modeling, and presumably the authors did not want to do it. Many of the responses to my review pointed to other related work by the authors. That's great, and their responses underscore the deep knowledge and insights that this team of authors have in the area of invasion ecology at the global scale. My additional suggestion would be to look through the manuscript again and make sure that the revised paper takes advantage of this previous work and points other readers (not just me as a reviewer) to those resources, findings, and insights. I think the paper could be slightly improved by drawing on these insights more fully.

We thank you for your insightful comments on our MS, which have greatly improved its clarity and depth. In response to your suggestion, we have carefully revisited the main text to ensure other relevant key references about genome-invasion relationships are cited. Specifically, we referenced the study by (te Beest *et al.* 2012), highlighting the role of polyploidy in plant invasion in lines 87-90: "This may potentially be because of polyploidy in the larger-genome species, as polyploidy not only results in a step change in genome size (at least initially) but can also generate heterozygosity, which might enhance competitive ability and increase the likelihood of successful invasion into new environments²⁶". Another key study summarizing the relationship between genome size and plant invasion (Suda *et al.* 2015) was cited in Tabel 1 (as a key reference for path 3) and lines 84-85: "Ultimately, these traits affect the habitat breadth and range size of a species and, consequently, its invasion potential^{14,17,24,25}". Moreover, we have added direct comparison to the recent study by (Pyšek *et al.* 2023), which examines direct genome-invasion relationship using comprehensive genome size data, in lines 218-221: "Moreover, while a recent study by Pyšek *et al.*¹⁷ acknowledges the direct significant impact of

genome size on plant naturalization and invasion, our SEMs at least partly unravel the underlying mechanisms of the large genome constraint effect by revealing two possible indirect pathways...".

Papers cited:

te Beest, M., Le Roux, J.J., Richardson, D.M., Brysting, A.K., Suda, J., Kubešová, M., *et al.* (2012). The more the better? the role of polyploidy in facilitating plant invasions. *Ann. Bot.*, 109, 19–45.

Pyšek, P., Lučanová, M., Dawson, W., Essl, F., Kreft, H., Leitch, I.J., *et al.* (2023). Small genome size and variation in ploidy levels support the naturalization of vascular plants but constrain their invasive spread. *New Phytol.*, 239, 2389–2403.

Suda, J., Meyerson, L.A., Leitch, I.J. & Pyšek, P. (2015). The hidden side of plant invasions: The role of genome size. *New Phytol.*, 205, 994–1007.